# Effective Balloon Pulmonary Angioplasty in a Patient with Chronic Thromboembolic Complications after Ventriculoatrial Shunt for Hydrocephalus in von Hippel–Lindau Disease

**DOI:** 10.3390/medicina58020185

**Published:** 2022-01-26

**Authors:** Piotr Gościniak, Michał Larysz, Leszek Sagan, Barbara Larysz, Anhelli Syrenicz, Marcin Kurzyna

**Affiliations:** 1Laboratory of Non-Invasive Cardiac Imaging, Independent Public Clinical Hospital Nr. 1, Pomeranian Medical University, 71-252 Szczecin, Poland; 2Endocrinology and Internal Diseases Department, Independent Public Clinical Hospital Nr. 1, Pomeranian Medical University, 71-252 Szczecin, Poland; klinendo@pum.edu.pl; 3Neurosurgery Department, Independent Public Clinical Hospital Nr. 1, Pomeranian Medical University, 71-252 Szczecin, Poland; michalarysz@gmail.com (M.L.); leszekm.sagan@gmail.com (L.S.); 4Cardiology Department, Maria Curie Skłodowska Province Hospital, 71-527 Szczecin, Poland; barbaralarysz@wp.pl; 5Department of Pulmonary Circulation, Thromboembolic Diseases and Cardiology, 05-400 Otwock, Poland

**Keywords:** von Hippel–Lindau, ventriculoatrial shunt, chronic thromboembolic pulmonary hypertension, balloon pulmonary angioplasty, multimodality imaging

## Abstract

Von Hippel–Lindau (VHL) disease along with chronic thromboembolic pulmonary hypertension (CTEPH) is a unique and unusual severe complication of ventriculoatrial (VA) shunt implantation in the treatment of hydrocephalus. To the best of our knowledge, this can be the first reported case of an effective treatment with balloon pulmonary angioplasty in a patient with VHL after VA shunt placement. The patient underwent six balloon pulmonary angioplasty procedures. All invasive procedures resulted in haemodynamic and functional improvement.

## 1. Introduction

Von Hippel–Lindau syndrome (VHL) is an inherited disease involving tumors that form in multiple organs. VHL-related tumors include hemangiomas: tumors commonly occurring in blood vessels of the brain, the spinal cord, and the retina.

Symptoms depend on the location of the tumor, its size, and increase in intracranial pressure [1]. The effect of a hemangioma mass may cause hydrocephalus, and creation of an artificial connection between the cerebral ventricle and the right atrium may be necessary [2]. Catheter malfunction and infection are serious complications associated with ventriculoatrial (VA) shunting of cerebrospinal fluid (CSF) [3]. However, chronic thromboembolic pulmonary hypertension (CTEPH) as a severe complication of the insertion of a VA shunt in the treatment of hydrocephalus is unique and uncommon [4]. 

We report a rare case of a 35-year-old patient with VHL disease who developed pulmonary hypertension as a severe complication of the insertion of a VA shunt to treat hydrocephalus. To the best of our knowledge, this can be the first case of an effective treatment with balloon pulmonary angioplasty in a patient suffering from VHL disease who received a VA shunt. 

## 2. Case Presentation

In 2014 the patient (then 31) was diagnosed with VHL disease presenting with typical multiple hemangiomas of the cerebellum and the spinal cord and elevated intracranial pressure due to obstructive hydrocephalus following a hemangioblastoma mass effect. In 2015 he underwent surgery for a posterior cranial cavity tumor, and a ventriculoatrial shunt was placed with the tip located in the right atrium. 

Two years after the surgery, the patient was admitted to the cardiology department with symptoms of heart failure, such as marked decrease in exercise tolerance, dyspnea, fatigue and ankle edema that increased over several months. The patient presented WHO functional class IV. The distance covered in the 6 min walk test was 109 m, with desaturation up to 81%. Plasma natriuretic peptide (NT-pro-BNP) levels were elevated to 1929 pg/mL (normal range <125 pg/mL). There were no laboratory, clinical, or chronic signs of infection at either the 2-year detailed post-surgery neurosurgical follow-up or on admission.

Echocardiography revealed right ventricular overload (Figure 1A,B) due to pulmonary hypertension. The distal end of the VA shunt was visible in the right atrium (Figure 1C), migrating into the right ventricle inflow tract (Figure 1D). A pulmonary perfusion scan (230 MBq of Tc99m intravenously) was performed to confirm the presence of pulmonary embolism, which showed multiple segmental and subsegmental perfusion deficits in both lungs (Figure 2A) [5]. Pulmonary angiogram by multidetector computed tomography (MDCT) showed significant mosaic attenuation in pulmonary fields of both lungs, as well as severe stenosis or even obstruction of the subsegmental pulmonary arteries (Figure 2B). In addition, fresh thrombi were detected. Right heart catheterisation followed by selective pulmonary angiography (PAG) showed a mean pulmonary artery pressure (mPAP) of 37 mmHg. Pulmonary artery wedge pressure was normal, and pulmonary vascular resistance was 13.9 Wood Units, indicating precapillary pulmonary hypertension (PH). Selective angiography confirmed chronic thromboembolic changes suggestive of CTEPH—segmental arteries and their branches were obstructed, amputated or distally altered on both sides (Figure 2C,D) [6]. The use of these multimodal methods was necessary to make a definitive diagnosis.

CTEPH was thus diagnosed and pharmacological treatment (sildenafil 25 mg tid.) was initiated along with anticoagulation therapy (rivaroxaban 20 mg). Due to the risk of hydrocephalus, complete removal of the catheter was not possible. Implantation into the peritoneal cavity was ineffective in previous surgeries due to peritoneal adhesions; hence, the treatment was abandoned. The distal end of the device was thus fixed at the entry of the superior vena cava (SVC) into the right atrium. A cardiac surgical consultation was sought but the patient was considered inoperable because of advanced heart failure, cerebral burden and likely complications associated with the use of extracorporeal circulation. Therefore, pulmonary balloon angioplasty (BPA) was planned. 

In 2018, the patient underwent six angioplasty procedures–three in the right lung and three in the left. A total of 45 segmental and subsegmental lesions were treated with balloon catheters with maximum diameters of 4 mm and 8 mm on the right and left side, respectively. (Figure 3A,B). The only complication observed was a vasovagal reaction after balloon inflation in the segmental branch. No reperfusion lung injury or hemoptysis was observed.

The procedure resulted in haemodynamic and functional, improvement, with a dramatic decrease in mPAP to 19 mmHg at 18-month follow-up (Figure 3C,D). The patient is now WHO functional class II, and tests have shown further improvement in RV clinical status and function (Table 1).

## 3. Discussion

Ventriculoatrial shunt placement is an effective method of draining CSF from the cerebral ventricle into the right atrium, and it significantly improves the survival of patients with hydrocephalus [7]. However, VA shunting is known to be a risk factor for CTEPH and is presently used only as an alternative to ventriculoperitoneal shunts [4,8].

The prevalence of PH among adult patients with VA shunts was estimated by Kluge et al. PH was determined by Doppler echocardiography in a significant proportion of patients (8%) with VA shunts [4].

Acute thrombus obstruction was observed in the superior vena cava (SVC) in the study by Al-Natour et al., who described an important case of SVC syndrome that developed as a consequence of VA shunting. The described procedure differed from our case because acute thrombus obstruction was diagnosed in the above study. Direct catheter thrombolysis with tissue plasminogen activator (tPA) was performed, followed by balloon angioplasty and SVC stenting [9].

Catheter infections may also be a cause of chronic non-thromboembolic PH. Extensive radiological investigation and normal ventilation/perfusion lung function test results allowed Amelot et al. to rule out CTEPH [10].

In a recent study Magrasi et al. proposed endovascular techniques when shunt removal was necessary. According to the authors, this less invasive method may be appropriate even in cases of overt endocarditis, atrial thrombi, and tight adherence to the endocardial wall [11].

The relative rarity of complications and long latency period between shunt placement and onset of symptoms in recipients of ventriculoatrial shunt may be the cause of potential misdiagnosis or delayed diagnosis. There are several theories as to why VA shunts may cause pulmonary hypertension, such as shunt infection or pulmonary endothelial response to CSF components, i.e., thromboplastin or serotonin, which may be responsible for the microembolisation process [8].

Currently, new treatment options have significantly improved the prognosis of patients with CTEPH [6,12]. The preferred treatment methodis pulmonary endarterectomy performed in centers experienced in management of this condition. Pulmonary endarterectomy is characterized by low post-operative mortality and good long-term survival rates. In patients ineligible for surgery, percutaneous balloon angioplasty is the increasingly preferred choice of treatment [13,14].

## 4. Conclusions

VA shunting as a treatment of hydrocephalus in VHL disease may be a source of clinically silent pulmonary embolism, resulting in the development of CTEPH. Patients who develop deterioration in physical function should immediately undergo echocardiography with assessment of pulmonary artery pressure. Our study strongly suggests that BPA and targeted therapy can be safe and effective options for treating CTEPH in VHL disease.

## Figures and Tables

**Figure 1 medicina-58-00185-f001:**
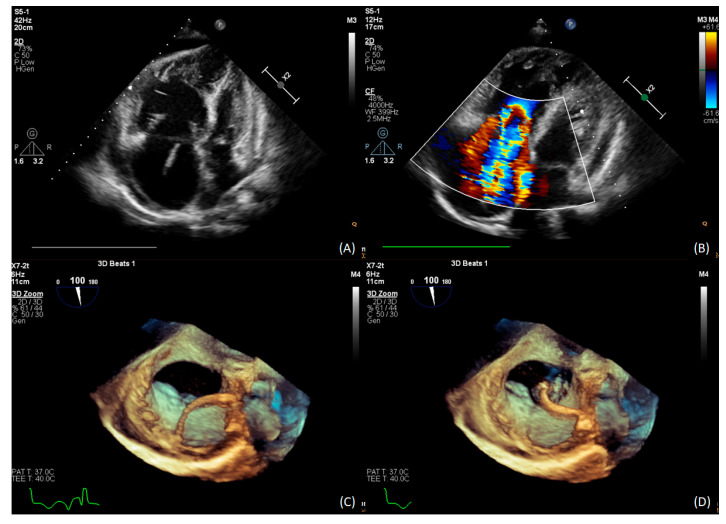
(**A**) TTE: severe dilatation of the right atrium and ventricle; (**B**) TTE: RV pressure overload with RVSP 55 mmHg; (**C**) 3D TEE distal end of the VA shunt in the right atrium; (**D**) 3D TEE the tip migration into the RVOT.

**Figure 2 medicina-58-00185-f002:**
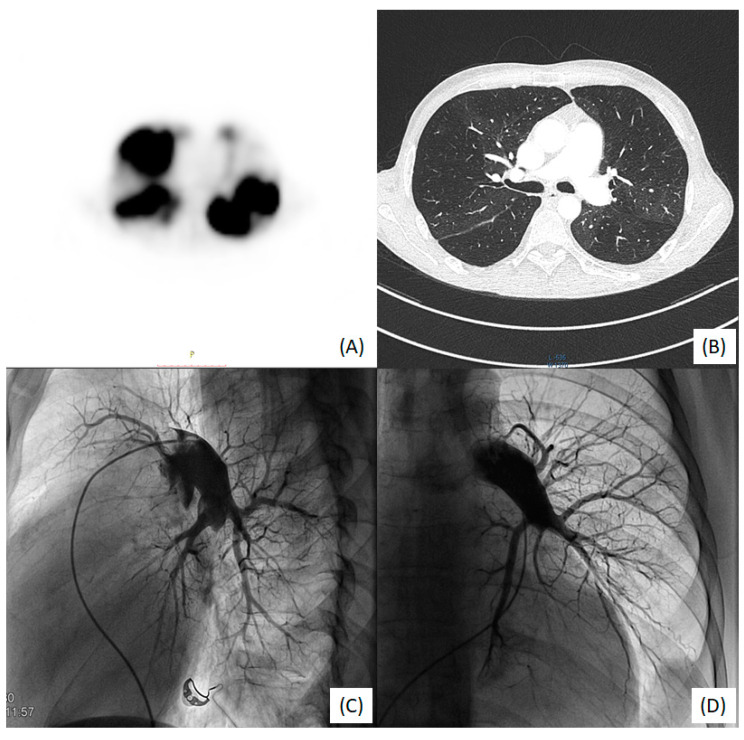
(**A**) V/Q scan: bilateral perfusion defects; (**B**) MDCT large bilateral stenoses, disparity and abridgment of in subsegmental vessel size; (**C**) right-side PAG: A9 and A10 arteries occluded and A4 amputated; (**D**) left-side PAG: amputation of the trunk A9/A10 and distal changes in A6 and A8.

**Figure 3 medicina-58-00185-f003:**
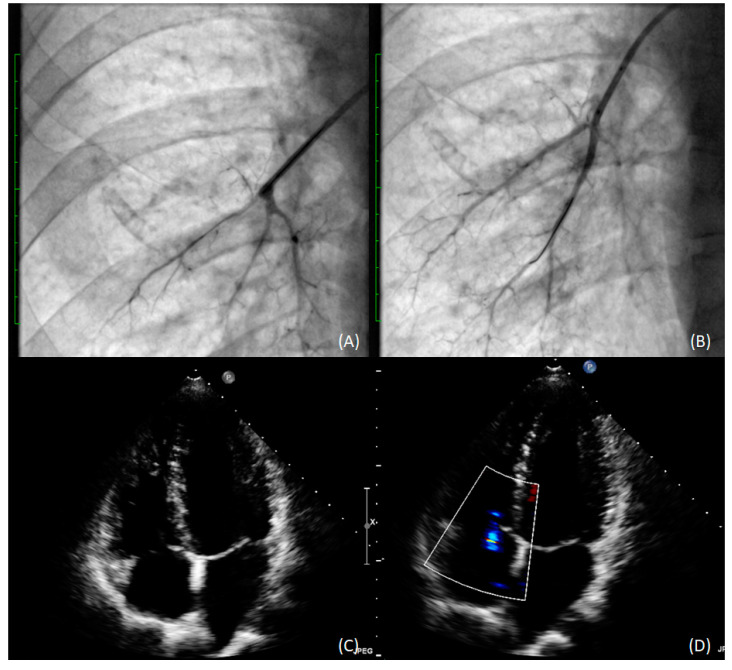
Subsegmental arteries in the lower lobe of the right lung before (**A**) and after (**B**) pulmonary angioplasty with 3.0 mm balloon catheter); (**C**) TEE with RV recovery; (**D**) TEE TRPG (31 mmHg) improvement.

**Table 1 medicina-58-00185-t001:** Clinical and haemodynamic assessment of the patient before treatment and at follow-up.

	Before Treatment	Follow-Up
WHO functional class	IV	II
NT-pro-BNP level (pg/mL)	1926	119
6MWT distance (m)	109	408
TRPG (mmHg)	40	31
RVSP (mmHg)	55	35
TAPSE (mm)	12	20
AcT (ms)	72	109
CI (l/min/m^2^)	1.3	2.45
PAP (mmHg) systolic/diastolic/mean	49/30/37	33/10/19
PVR (Wood Units)	13.97	2.19
RAP mean	23	2

AcT pulmonary velocity acceleration time; CI cardiac index; PAP pulmonary artery pressure; PVR pulmonary vascular resistance; RAP right atrial pressure; RVSP right ventricular systolic pressure; TAPSE tricuspid annular plane systolic excursion; TRPG tricuspid regurgitation pressure gradient.

## Data Availability

Data available upon request from M.K.

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
