# Peer review of "Effective Balloon Pulmonary Angioplasty in a Patient with Chronic Thromboembolic Complications after Ventriculoatrial Shunt for Hydrocephalus in von Hippel–Lindau Disease"

_medicina, 2022, doi:10.3390/medicina58020185_

Round 1

Reviewer 1 Report

Effective balloon pulmonary angioplasty in a patient with chronic thromboembolic complications from a ventriculoatrial shunt for hydrocephalus in von Hippel-Lindau disease

Line 19-21 and line 29-31

“To the best of our knowledge, this is the 29 first described case of treatment with effective balloon pulmonary angioplasty in a patient 30 with VHL disease after receiving a VA shunt”.

This is the second. The first but without a Von Hippel-Lindau disease:

"Al-Natour MS, Entezami P, Nazzal MM, Casabianca AB, Assaly R, Riley K, Gaudin D. Superior vena cava syndrome with retropharyngeal edema as a complication of ventriculoatrial shunt. Clin Case Rep. 2015 Oct;3(10):777-80. doi: 10.1002/ccr3.331".

Please cite and comment this article.

Line 38-39

You must report the age in which she has been submitted to ventriculoatrial shunt and remove the citation “2” (Varshney 2017) since it is not pertinent with the clinical description.

Line 44-48

In this section you lack any clinical information concerning possible previous or concurring infections which can cause thromboembolic complications in atrial shunts. Serological data as Polymerase chain reaction (PCR) must be reported because its alteration explain the causative factor for atrial and pulmonary embolism.

Line 49-63

This text result from the “Guidelines for Authors”. Please remove.

The same on the line 84-86

Line 90

“Removal of the catheter was planned”

Please, this is a very important issue. How the patient was treated according the atrial shunt? It was temporarily removed and substituted in another location, i.e. in the peritoneal cavity? She had hydrocephalus? You must report.

This paper is interesting from the neurosurgical point of view, but needs some changes.

Author Response

Rebuttal Letter

We thank the Editors and Reviewers for their time and effort.  In the following, we address the raised issues point-by-point.

Reviewer 1

#

Line 19-21 and line 29-31

“To the best of our knowledge, this is the 29 first described case of treatment with effective balloon pulmonary angioplasty in a patient 30 with VHL disease after receiving a VA shunt”.

This is the second. The first but without a Von Hippel-Lindau disease:

"Al-Natour MS, Entezami P, Nazzal MM, Casabianca AB, Assaly R, Riley K, Gaudin D. Superior vena cava syndrome with retropharyngeal edema as a complication of ventriculoatrial shunt. Clin Case Rep. 2015 Oct;3(10):777-80. doi: 10.1002/ccr3.331".

Reply: This is a very interesting and similar case, as recommended, we recited the article in the discussion. However, we believe that the patient's underlying disease and treatment are different.

#

Line 38-39

You must report the age in which she has been submitted to ventriculoatrial shunt and remove the citation “2” (Varshney 2017) since it is not pertinent with the clinical description.

Reply: details of relevant past medical history have been supplemented.
Citation was removed.

#

Line 44-48

In this section you lack any clinical information concerning possible previous or concurring infections which can cause thromboembolic complications in atrial shunts. Serological data as Polymerase chain reaction (PCR) must be reported because its alteration explain the causative factor for atrial and pulmonary embolism.

Reply: neither during the 2-year detailed neurosurgical follow-up nor on admission were laboratory, clinical or chronic signs of infection observed.

#

Line 49-63

This text result from the “Guidelines for Authors”. Please remove.

Reply: corrected according to "Guidelines for Authors"

#

The same on the line 84-86

Reply: corrected according to "Guidelines for Authors"

#

Line 90

“Removal of the catheter was planned”

Please, this is a very important issue. How the patient was treated according the atrial shunt? It was temporarily removed and substituted in another location, i.e. in the peritoneal cavity? She had hydrocephalus? You must report.

Reply: Due to the risk of hydrocephalus, it was not possible to completely remove the catheter. Implantation to the peritoneal cavity was inefficient in previous surgeries due to copious peritoneal adhesions therefore it was abundant during current treatment. Eventually the distal end of the device was fixed at the outlet of the superior vena cava to the right atrium.

Reviewer 2 Report

Dear Authors,

The main contribution of this case is its high originality. Balloon pulmonary angioplasty in patients with chronic thromboembolic pulmonary hypertension is well described in the literature, but never as a long-term complication of VA shunt insertion in a patient with VHL disease. Methodological design is accurate: investigations are described with sufficient details and suitable images. Figures and table properly show the data and are easy to understand. Results are well supported by literature references and follow-up data.

“Medical History” and “Investigations” include inappropriate writing that you have to remove from 49 to 63 and from 84 to 86.

Introduction should be improved. In this section you should give more information about VHL disease and its complications, including hydrocephalus shunt treatments and its complications. More information about patient anamnesis should be given too: you should specify when the shunt was implanted and how long after the patient manifested heart failure symptoms.

Most references are current and appropriated, except the second and the fourth. In 41 you described patient medical history and no need to cite an article. In 78, patient’s mPAP is reported without adding further details, so self-citation is inappropriate.

Discussion should be improved: VHL disease, hydrocephalus and shunt treatments should be deepened because they represent the novelty of your paper. You should report and discuss similar cases.

In Conclusion you should increase focus on key-points and give practical advice on diagnosis and therapy.

Author Response

Rebuttal Letter

We thank the Editors and Reviewers for their time and effort.  In the following, we address the raised issues point-by-point.

Reviewer 2

#

“Medical History” and “Investigations” include inappropriate writing that you have to remove from 49 to 63 and from 84 to 86.

Reply: corrected according to "Guidelines for Authors"

#

Introduction should be improved. In this section you should give more information about VHL disease and its complications, including hydrocephalus shunt treatments and its complications

Reply: corrected as recommended

#

More information about patient anamnesis should be given too: you should specify when the shunt was implanted and how long after the patient manifested heart failure symptoms.

Reply: details of relevant past medical history have been supplemented.

#

Most references are current and appropriated, except the second and the fourth. In 41 you described patient medical history and no need to cite an article. In 78, patient’s mPAP is reported without adding further details, so self-citation is inappropriate.

Reply: corrected as recommended

#

Discussion should be improved: VHL disease, hydrocephalus and shunt treatments should be deepened because they represent the novelty of your paper. You should report and discuss similar cases.

Reply: corrected as recommended

Round 2

Reviewer 1 Report

Now all the clinical informations are detailed and can allow better understanding to the reader.

Author Response

Rebuttal Letter

In revised manuscript, we have carefully considered reviewers’ comments and suggestions.

We reply to each comment in point by point fashion.

#

Extensive editing of English language and style required

Reply: the entire text corrected as recommended

#

From line 27 to 30 I suggest a major rewrite.

Reply: corrected as recommended

#

In line 33 you should explain to the reader what “Pudenz valve” is

Reply:

#

Line 42-46 should be moved into “case presentation”, before you described patient’s diagnostic tests.

Reply: corrected as recommended

#

You should not put time references in brackets: integrate them into the text. 

Reply: corrected as recommended

#

If you specify in line 54 "after two years from surgery…", in line 55 “2017” is not necessary.

Reply: corrected as recommended

#

In line 77 you should specify sildenafil dosage as you did for rivaroxaban.

Reply: corrected as recommended

#

From 102 to 105 you should reorganize the concepts, putting the reference at the end, as you did for other discussed cases.

Reply: corrected as recommended

#

Conclusions should be written as a single text, without a subdivision.

Reply: corrected as recommended

Reviewer 2 Report

Dear Authors,   The manuscript has been improved from the last version, but some changes are yet necessary to published it.  Introduction:  - From line 27 to 30 I suggest a major rewrite. - In line 33 you should explain to the reader what “Pudenz valve” is. - Line 42-46 should be moved into “case presentation”, before you described patient’s diagnostic tests. Case presentation: - You should not put time references in brackets: integrate them into the text. If you specify in line 54 "after two years from surgery…", in line 55 “2017” is not necessary. - In line 77 you should specify sildenafil dosage as you did for rivaroxaban. Discussion:  - From 102 to 105 you should reorganize the concepts, putting the reference at the end, as you did for other discussed cases. Conclusions should be written as a single text, without a subdivision.    Best regards

Author Response

(The authors gave the same response as above.)
